# Evaluating 3D-printed models of coronary anomalies: a survey among clinicians and researchers at a university hospital in the UK

Matthew Lee,[1] Sarah Moharem-Elgamal,[2,3] Rylan Beckingham,[1] Mark Hamilton,[3] Nathan Manghat,[3] Elena Giulia Milano,[4,5] Chiara Bucciarelli-Ducci,[1,3] Massimo Caputo,[1,3] Giovanni Biglino [1,3,5]

[1]Bristol Medical School, University of Bristol, Bristol, UK
[2]National Heart Institute, Giza, Egypt
[3]University Hospitals Bristol, NHS Foundation Trust, Bristol, UK
[4]Division of Cardiology, Department of Medicine, University of Verona, Verona, Italy
[5]Cardiorespiratory Division, Great Ormond Street Hospital for Children, NHS Foundation Trust, London, UK

**Correspondence to**
Dr Giovanni Biglino;
g.biglino@bristol.ac.uk

## ABSTRACT

**Objective** To evaluate the feasibility of three-dimensional (3D) printing models of coronary artery anomalies based on cardiac CT data and explore their potential for clinical applications.

**Design** Cardiac CT datasets of patients with various coronary artery anomalies (n=8) were retrospectively reviewed and processed, reconstructing detailed 3D models to be printed in-house with a desktop 3D printer (Form 2, Formlabs) using white resin.

**Setting** A University Hospital (division of cardiology) in the UK.

**Participants** The CT scans, first and then 3D-printed models were presented to groups of clinicians (n=8) and cardiovascular researchers (n=9).

**Intervention** Participants were asked to assess different features of the 3D models and to rate the models' overall potential usefulness.

**Outcome measures** Models were rated according to clarity of anatomical detail, insight into the coronary abnormality, overall perceived usefulness and comparison to CT scans. Assessment of model characteristics used Likert-type questions (5-point scale from 'strongly disagree' to 'strongly agree') or a 10-point rating (from 0, lowest, to 10, highest). The questionnaire included a feedback form summarising overall usefulness. Participants' imaging experience (in a number of years) was also recorded.

**Results** All models were reconstructed and printed successfully, with accurate details showing coronary anatomy (eg, anomalous coronary artery, coronary roofing or coronary aneurysm in a patient with Kawasaki syndrome). All clinicians and researchers provided feedback, with both groups finding the models helpful in displaying coronary artery anatomy and abnormalities, and complementary to viewing 3D CT scans. The clinicians' group, who had substantially more imaging expertise, provided more enthusiastic ratings in terms of models' clarity, usefulness and future use on average.

**Conclusions** 3D-printed heart models can be feasibly used to recreate coronary artery anatomy and enhance understanding of coronary abnormalities. Future studies can evaluate their cost-effectiveness, as well as potentially explore other printing techniques and materials.

### Strengths and limitations of this study

► The study gathers evidence towards clinical and research uses of three-dimensional (3D) cardiac models through involvement of both clinicians and researchers as key stakeholders.

► Feedback obtained from stakeholders provided qualitative and quantitative information on the models' usefulness, clarity of anatomical visualisation and insight into the abnormality.

► Stakeholders were shown CT imaging data as well as 3D cardiac models, allowing for comparisons to be drawn on their effectiveness.

► Selected cases were chosen from a database of patients having coronary CT; a general population incidence of these anomalies is unable to be estimated.

► The study is limited by the small number of participants (n=17), and further studies looking at other anomalies may be needed to extrapolate our results to varying classifications of the anomalies.

## INTRODUCTION

Three-dimensional (3D) printing offers unique advantages in medicine, with great potential for personalisation in the face of the wide variation of anatomy, morphology and disease across individuals. This technology has seen rapid advancement in its development and applications, with evidence beginning to emerge in support of its effectiveness and clinical value.[1] Benefit has been found in pharmaceutical and interventional research, printing of anatomical models, tissue fabrication, prosthetics and implants.[2] The use of 3D models in teaching, clinical consultations and preoperative planning has also been assessed.[1 3 4] The technology has seen a rapid rate of adoption in congenital cardiology, with models aiding understanding of complex anatomical structures and accurately delineating anatomical morphology to make informed management decisions.[5]

Areas of positive early experience include preoperative planning, functional flow models assessing aortic valve dysfunction, device innovation and teaching.[6 7] Enhanced understanding of complex anatomical spatial relationships through patient-personalised 3D models could lead to improving surgical outcomes.[3]

Coronary anatomy has serious implications on cardiac premature morbidity and mortality. Congenital coronary artery anomalies (CAAs) present a wide variation in morphology, and are prevalent in approximately 1% of the population.[8] They are usually asymptomatic and encountered as an incidental finding during coronary angiography.[9] However, CAAs are the second most common cause of sudden cardiac death in young athletes,[10] for example, representing the predominant (61%) identifiable cardiac abnormality in a study looking at non-traumatic sudden deaths of young healthy military recruits.[11]

Acquired, non-atherosclerotic CAAs can be seen in Kawasaki disease, a systemic inflammatory condition of medium-sized vessels, with a predilection for the coronary arteries.[12] Coronary artery aneurysms are the most common complication of Kawasaki coronary arteritis. Coronary imaging in this population is important for serial follow-up, management and prognosis. The detection of coronary complications in Kawasaki disease can be challenging, as it usually presents in children, who have small vessel size and increased heart rates.

Coronary CT angiography is currently the most widely used non-invasive imaging modality for investigating CAA, offering detailed visualisation of the anatomy and coronary arteries with higher sensitivity than invasive coronary angiography.[13 14] A single widely accepted CAA classification scheme is lacking, despite attempts by various groups to sort them by properties including clinical symptoms, anatomical appearance, origin and risk.[9 10 15 16] The use of CT imaging within 3D printing in medicine has been widely established.[17] This study, thus, aims to explore the use of 3D-printed models for viewing and investigating CAAs, evaluating the feasibility of its use both clinically and in an investigative capacity. Two key stakeholders were engaged: (1) clinicians, who diagnose CAAs or are involved in management decisions and (2) non-medically qualified cardiovascular researchers, who may benefit from clearer viewing of coronary arteries for research purposes.

## MATERIALS AND METHODS
### Case selection
Two cardiac radiology consultants with >15 years experience in cardiac CT (NM, MH) selected n=7 cases from our centre's database of roughly 2000 patients having coronary CT over the last 10 years. An additional case with normal coronary anatomy was included ('control case'). The criteria for selecting the cases were: range of CAA.
1. Available CT coronary angiogram.
2. Consent for research use of images.

Cases were purposely selected to best represent a spectrum of CAAs at different incidences and timelines. All scans were displayed on a 512×512 matrix (table 1). An illustrated diagram of the study design is shown in figure 1. Considering the focus of the study on printing feasibility and model evaluation, the study is not considered Clinical Research as defined in the UK Policy Framework for Health and Social Care Research, and Health Research Authority (HRA) and National Health System (NHS) Research Ethics Committee (REC) approval were

| Table 1 | Acquisition and clinical information of patients and their scans for each case | | | | | |
|---|---|---|---|---|---|---|
| | **Acquisition information** | | | **Clinical information** | | |
| **Cases** | **Scanner** | **Slice thickness (mm)** | **Detectors** | **Age (year)** | **Gender** | **Pathology** |
| Case 1 | Toshiba Aquilion ONE | 0.50 | 320 | 39 | M | Normal coronary anatomy |
| Case 2 | Toshiba Aquilion ONE | 0.50 | 320 | 13 | F | Multiple anomalous coronary arteries |
| Case 3 | Toshiba Aquilion ONE | 0.50 | 320 | 10 | F | Coronary fistula |
| Case 4 | Toshiba Aquilion ONE | 0.50 | 320 | 36 | M | Myocardial bridging of coronary artery |
| Case 5 | SOMATOM Definition AS+ | 0.75 | 128 | 48 | M | Tetralogy of Fallot; LCX and LAD come off separately from aorta |
| Case 6 | Toshiba Aquilion ONE | 0.50 | 320 | 18 | M | Transposition of great arteries; Abnormal circumflex artery |
| Case 7 | SOMATOM Definition AS+ | 0.75 | 128 | 52 | F | Kawasaki's disease; left main stem coronary artery aneurysm |
| Case 8 | Toshiba Aquilion ONE | 0.50 | 320 | 56 | F | Anomalous left coronary artery from pulmonary artery |

LAD, left anterior descending artery; LCX, left circumflex artery.

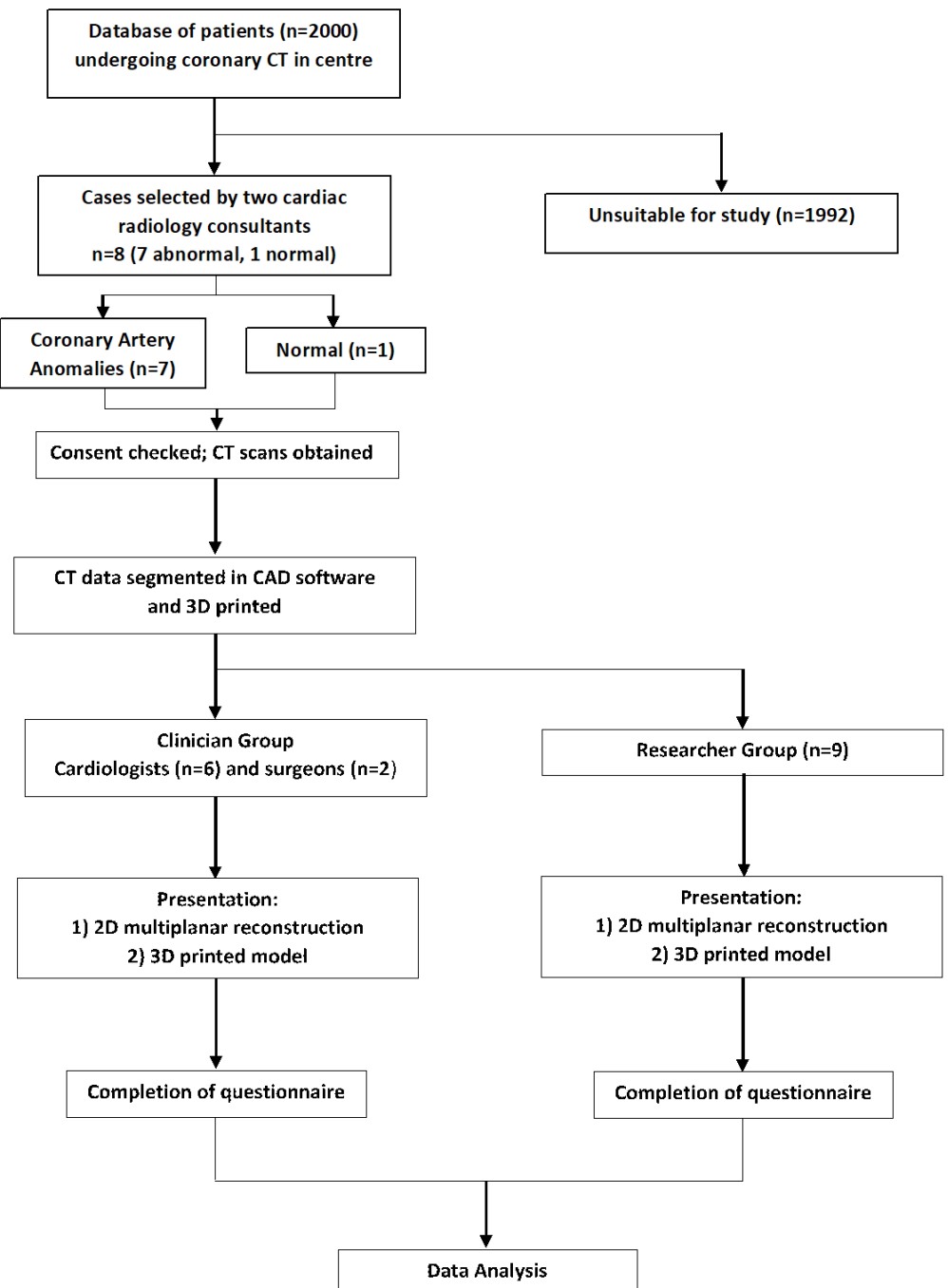

**Figure 1** Illustrated diagram of study design. 2D, two-dimensional; 3D, three-dimensional; CAD, computer aided design.

not deemed necessary by the local Research & Innovation committee.

### CT protocol

Cardiac CT scans were performed on a 320-multidetector row CT system (Toshiba Aquilion ONE) or 128-slice cardiac CT (Siemens Medical Solutions, Erlangen, Germany). Patients received 20–60 mL of iodinated contrast (calculated at 20 mg iodine/kg/second for 14–20 s), followed by 10–40 mL of normal saline. Scan timing by bolus tracking

(Toshiba) and test bolus (Siemens) were performed with region of interest in the ascending aorta.

### 3D printing

The anonymised scans were processed with commercial software (Mimics Research 19.0, Materialise NV, Leuven, Belgium) reconstructing the 3D heart model following steps of layer masking, segmentation and region growing.[18] Each scan was assessed by a medical student (ML) with the guidance of a cardiologist with

10 years in cardiac CT (SM-E) to decide the best way to display the CAA, purposefully removing certain structures to reveal the coronary arteries and abnormality more clearly. The decision to remove parts of the cardiovascular anatomy was made on the basis that the direct line of sight to view and follow the coronary arteries was not obstructed. Models were 1:1 in size. While the accuracy of the reconstruction protocol has been previously demonstrated,[18] the resulting 3D reconstructions were also verified visually by the cardiologist against the original CT dataset.

Models were smoothed (3-matic Research 11.0, Materialise) and exported as stereolithography (STL) files. The STL files were imported into the 3D printing software (PreForm 2.10.3, Formlabs, Somerville, Massachusetts, USA) where orientation, scaffolding and print layout were set before printing the model. The printer (Formlabs Form 2 SLA) was available in-house. Models were all printed in white resin. On completion of printing, the models were submerged in propanol for 25–30 min, then dried in a fume cupboard for an hour. The scaffolding was removed manually. Models were then assessed for quality by a biomedical engineer with 7 years experience of 3D printing (GB) and checked for correct anatomical representation with respect to the CT scan by the cardiologist.

### Models presentation and feedback

A questionnaire was devised to collect feedback on the models' usefulness and clarity from two groups of stakeholders (table 2). Cardiovascular researchers (n=9) had a background in cardiovascular science but had little or no experience of imaging. Clinicians included cardiac surgeons (n=2) and cardiologists (n=6, of which n=3 with extensive experience in cardiac imaging). Cardiologists and radiologists directly involved in patients' selection and model preparation did not take part in the survey.

A presentation was given to the researchers, where all models and their respective CT scans were shown one by one, including basic clinical information for each case (eg, age, gender, diagnosis, medical history). Screenshots of the CT scans showing coronary anatomy were displayed, followed by introduction of the model which was then passed around and examined. A similar drop-in session was organised for the clinicians to accommodate their less flexible schedules, consisting of the same presentation and feedback questionnaire. The presentation lasted approximately 1 hour. The researcher administering the survey had no prior relationship with participants.

The questionnaire focused on assessing clarity of anatomical visualisation, insight into the abnormality, overall usefulness and comparison to CT scans. Assessment of model characteristics used Likert-type questions on a 5-point scale (from 'strongly disagree' to 'strongly agree') or a 10-point rating (from 0, lowest score, to 10, highest score). The questionnaire also included a final feedback form summarising overall usefulness and asking for participants' imaging experience (in a number of years). One question regarding usefulness to future practice/applications was modified slightly to suit either research or clinical practice (ie, usefulness 'as part of your clinical practice'/'as part of your research'). Stakeholders were asked to fill in a feedback form following presentation of each case and a final feedback form.

### Data analysis

Analysis of questionnaire responses was carried out in Stata (V.13, StataCorp). Responses are presented as counts or mean±SD as appropriate. Unpaired comparisons between the two study groups (clinicians vs researchers) were carried out using Wilcoxon rank-sum test. Open-ended feedback was qualitatively analysed for dominant themes.

### Patient and public involvement

The study involved key stakeholders in the technology (ie, clinicians and researchers), according to a model of social construction of technology. In this instance, patients were not involved as beyond the specific focus of the study, despite being key stakeholders in the technology.

**Table 2** Purposes for inclusion of various specialties within clinician group

| Party | Purpose for inclusion in feedback | How would three-dimensional (3D) modelling be useful to this party? |
|---|---|---|
| Researcher | Strong cardiovascular background but likely limited experience with cardiac imaging and congenital heart disease. | Appreciation of anatomy across different congenital coronary artery diseases; potential use of models in future research. |
| Cardiac surgeon | Experienced with cardiac anatomy, pathology and imaging; specific focus on potential use of models for preoperative assessment. | Understanding specific visuospatial anatomy abnormality; preoperative planning and practice. |
| Radiologist/imaging expert | Extremely experienced with imaging; assessing whether 3D models improve/complement appreciation from CT alone. | Visuospatial appreciation of abnormality in addition to what they see on scans. |
| Cardiologist | Experienced with cardiac anatomy, pathology and imaging; assessing whether models can improve/complement understanding. | Use for complex cases with difficult coronary anatomy; support management decisions based on measurable dimensions. |

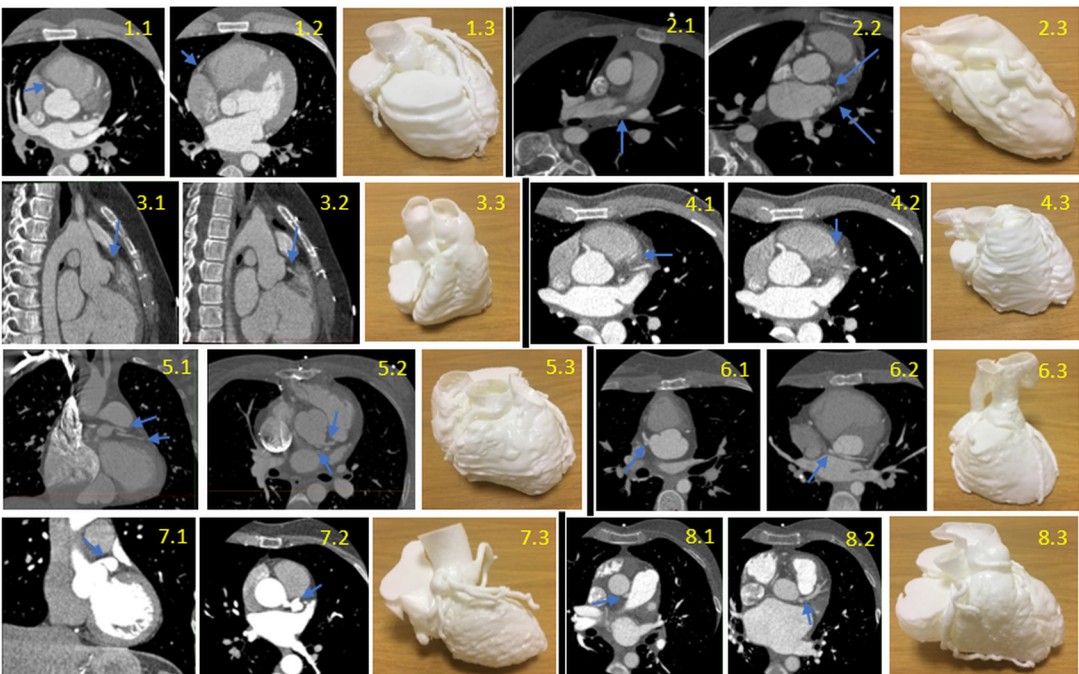

**Figure 2** CT Scans and models of each case, with the blue arrows pointing to the abnormality on the CT. Case 1 (first row left 1.1–1.3): normal coronary anatomy; axial view progressing inferiorly (1.1, 1.2). Case 2 (first row right 2.1–2.3): multiple anomalous coronary arteries; axial view progressing inferiorly (2.1, 2.2). Case 3 (second row left 3.1–3.3): coronary fistula; sagittal view progressing right (3.1, 3.2). Case 4 (secondd row right 4.1–4.3): myocardial bridging; axial view progressing inferiorly (4.1, 4.2). Case 5 (third row left 5.1–5.3): separate left circumflex (LCX) and left anterior descending (LAD) arteries from the aorta, tetralogy of Fallot; coronal view (5.1) and axial view (5.2). Case 6 (third row right 6.1–6.3): transposition of great arteries with abnormal circumflex artery; axial view progressing inferiorly (6.1, 6.2). Case 7 (fourth row left 7.1–7.3): Kawasaki's disease with left main stem coronary artery aneurysm; coronal view (7.1), axial view (7.2). Case 8 (fourth row right 8.1–8.3): anomalous left coronary artery from pulmonary artery; axial view progressing inferiorly (8.1, 8.2).

## RESULTS

The CT scans and models of each case are shown (figure 2). CT scans were of adequate quality for segmentation. All models were reconstructed and printed successfully. Details of each printed model are reported (table 3). All questionnaires were completed successfully apart from one incomplete in the researchers group.

Coronary artery anatomy was interpreted clearly by both groups and CAAs were identified clearly on the models by clinicians (8.9±0.8/10) and researchers (7.4±0.3/10). The abnormality was only missed in 2% of cases overall. Clinicians tended to rate the models slightly higher than researchers on average in terms of clarity of anatomy (8.9±0.8 vs 7.5±0.4, p=0.02), abnormality (9.0±0.7 vs 7.5±0.5, p=0.03) and usefulness (9.1±0.7 vs 7.7±0.5, p=0.03).

Looking at both groups combined and accounting for any incomplete answer, in 98% of cases (126/130 cases), models enhanced the awareness of coronary anatomy and abnormality. Clinicians found the models more useful overviewing the CT scans alone (4.5±0.3/5), as did researchers (4.2±0.3/5). Almost all participants (16/17) stated that the models would be useful in any future work involving CAAs. The dominant themes emerging from

| Table 3 | Details of each printed model | | | |
|---|---|---|---|---|
| Case | Reconstruction time (hours) | Resin volume (mL) | Layers (n) | Print duration |
| Case 1 | 4.5 | 139.2 | 3202 | 22 hours 51 min |
| Case 2 | 6 | 88.36 | 2498 | 14 hours 24 min |
| Case 3 | 5 | 81.71 | 2282 | 12 hours 43 min |
| Case 4 | 9 | 130.6 | 2626 | 29 hours 30 min |
| Case 5 | 7 | 214.3 | 3225 | 33 hours 05 min |
| Case 6 | 6.5 | 107.3 | 2683 | 17 hours 45 min |
| Case 7 | 4.5 | 76.70 | 2479 | 13 hours 45 min |
| Case 8 | 4 | 154.0 | 3334 | 25 hours 01 min |

**Table 4** Dominant themes emerging from comments received by clinicians and researchers

| Clinicians' comments | n | Researchers' comments | n |
|---|---|---|---|
| Enhanced visuospatial awareness using the model | 5 | Model was easier to understand after some guidance | 4 |
| It was useful to remove parts of structures to follow the coronary arteries | 3 | Removal of certain parts useful to give focus on the part of interest | 2 |
| Models would be effective for training of postgraduates and medical students | 3 | Use of colouring to delineate coronary arteries and abnormalities would make model more effective | 2 |
| Anomalous left coronary artery from pulmonary artery (case 8) was difficult to identify on the CT scans, but the anatomy was clear on the model | 3 | Understanding of anatomy improved as more models looked at | 1 |
| Use of colouring to delineate coronary arteries and abnormalities would make model more effective | 2 | | |

clinicians' and researchers' comments are reported in table 4. Evaluation of model usefulness is reported in detail in table 5.

## Clinicians

All clinicians had at least moderate expertise to view and interpret scans, with five of them having advanced imaging expertise and working with scans daily. Generally, clinicians found the coronary arteries easy to follow and reported to easily identify the anatomy and abnormality. In 89% of all cases, they agreed that the 3D model was more effective for their understanding of the abnormality than just looking at the CT scan, with 74% stating they 'strongly agreed'. Clinicians rated the models for case 6 (transposition of great arteries [TGA] with abnormal circumflex artery) and case 8 (anomalous left coronary artery from pulmonary artery [ALCAPA]) the clearest and most effective. All clinicians agreed that models would be useful in their future work with CAAs and suggested use for:

▸ Junior and senior radiologists, to demonstrate the diagnostic CT anatomy to clinicians.
▸ Explaining conditions to patients.
▸ Surgeons, during preoperative planning.

## Researchers

In most cases (seven out of nine) researchers had only basic imaging expertise or less, with five of them having no imaging expertise. In general, researchers could interpret the model well, but their scores of clarity of anatomy and clarity of abnormality were slightly less enthusiastic than clinicians' scores. In 97% of cases, researchers agreed that viewing the 3D model was more effective for their understanding than just looking at the CT scan, with 17% 'strongly agreeing' it was more effective. Each researcher reported that in an average of 69% of cases the models enhanced their anatomical awareness, and that the course of coronary arteries was easy to follow. Researchers gave the highest ratings to case 6 (TGA), case 7 (Kawasaki's disease coronary aneurysm, figure 3) and case 8 (ALCAPA, figure 4). The open-ended feedback revealed that researchers found the models a bit more

difficult to interpret, but eight out of nine agreed they would be useful in any future work studying CAAs.

Researchers interestingly remarked on the potential for clinical applications of the models and particularly mentioned:

▸ Surgeons in making presurgical decisions.
▸ Clinicians to make management decisions.
▸ Helping patients and families understand disease.

## DISCUSSION

Considering the complexity of CAAs, patient-specific models can provide enhanced visuospatial appreciation of the defect in relation to other anatomical structures and increase the user's understanding of the anatomy, as shown in this study. Enhanced anatomical appreciation has been proven to lead to better preoperative planning and management decisions within a clinical setting.[19] This appreciation of relational structures is particularly vital in the clinical picture of CAAs, where evaluation of the nature, location and course to determine degree of abnormality of the CAA and risk of harm to the patient can significantly alter management decisions.[9 10 20]

3D printing has been used to reconstruct and print the coronary arterial tree[21] and it has been demonstrated that 3D printing models were effective in visualising the coronary arteries.[5] At present, only two case reports present 3D-printed models for the evaluation of coronary fistulae.[22 23] In these cases, the 3D-printed model anecdotally added value in management decisions. However, no studies have been carried out to evaluate clinical use of 3D-printed models for coronary arteries and CAAs.

Currently, CAAs are investigated primarily by CT angiography, which has shown to have an excellent detection rate and is fast becoming the imaging modality of choice.[13 14] Although the CT scans may adequately delineate coronary anatomy, our study showed that the complementary use of 3D-printed models was useful for viewing coronary anatomy and anomalies by researchers and clinicians with imaging and cardiovascular background, and was considered effective for increasing their understanding of the abnormality over simply viewing the CT

**Table 5** Evaluation of model usefulness

| Case | Diagnosis | Clarity of anatomy average (out of 10) | | | Clarity of abnormality average (out of 10) | | | Usefulness average (out of 10) | | | Effectiveness over CT scan (from 1=strongly disagree to 5=strongly agree) | | |
|---|---|---|---|---|---|---|---|---|---|---|---|---|---|
| | | Clinicians | Researchers | P value | Clinicians | Researchers | P value | Clinicians | Researchers | P value | Clinicians | Researchers | P value |
| 2 | Multiple anomalous coronaries | 8.9±1.5 | 6.9±1.5 | 0.021* | 9.4±1.4 | 6.9±1.5 | 0.006* | 9.2±1.4 | 6.7±1.6 | 0.009* | 4.6±0.7 | 3.9±0.6 | 0.036* |
| 3 | Coronary fistula | 7.5±3.0 | 7.0±1.1 | 0.520 | 7.9±2.7 | 6.5±1.8 | 0.108 | 8.0±2.8 | 7.1±1.1 | 0.142 | 4.1±0.8 | 4.0±0 | 0.607 |
| 4 | Bridging | 8.2±2.8 | 7.4±0.8 | 0.072 | 8.1±2.7 | 7.4±0.7 | 0.068 | 8.4±2.7 | 7.3±0.9 | 0.033* | 4.4±1.1 | 4.3±0.5 | 0.365 |
| 5 | Separate LCX and LAD from aorta | 9.2±1.4 | 7.2±1.2 | 0.010* | 9.2±1.4 | 7.0±1.8 | 0.010* | 9.1±1.4 | 7.0±1.6 | 0.013 | 4.2±0.7 | 3.8±0.7 | 0.158 |
| 6 | TGA, abnormal circumflex | 9.5±0.7 | 7.5±0.9 | 0.002* | 9.4±0.9 | 7.5±0.8 | 0.003* | 9.5±0.8 | 7.6±0.8 | 0.003* | 4.8±0.5 | 4.4±0.5 | 0.221 |
| 7 | Kawasaki's aneurysm | 9.2±1.4 | 8.1±1.1 | 0.041* | 9.4±1.4 | 8.2±1.0 | 0.027* | 9.2±1.4 | 8.4±0.9 | 0.055 | 4.6±0.7 | 4.4±0.5 | 0.346 |
| 8 | ALCAPA | 10.0±0 | 7.8±0.7 | <0.001* | 9.9±0.3 | 7.8±1.0 | <0.001* | 9.9±0.3 | 8.1±0.6 | <0.001* | 5.0±0 | 4.3±0.5 | 0.005* |

\* indicates statistical significance.

ALCAPA, anomalous left coronary artery from pulmonary artery; LAD, left anterior descending; LCX, left circumflex; TGA, transposition of the great arteries.

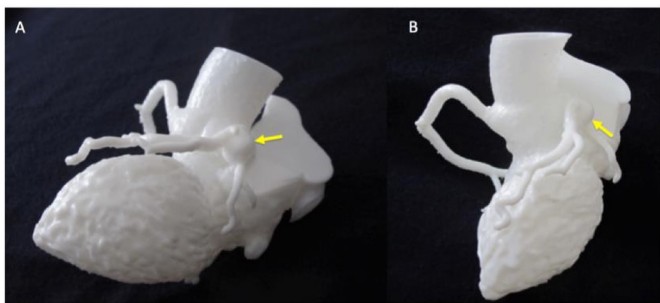

**Figure 3** Detail of Kawasaki model for appreciation of coronary aneurysm (indicated by yellow arrows), two different views (A and B).

scans. A randomised controlled trial by Li *et al* recently drew similar conclusions in comparing CT scans versus 3D modalities for teaching purposes. In this case, two-dimensional (2D) CT images, monochromatic 3D virtual models and monochromatic 3D-printed models were used for teaching cervical and thoracic spinal anatomy and fractures. A 10-mark post-teaching examination showed that students with teaching using 3D-virtual and 3D-printed models scored higher than students in the CT group, demonstrating efficacy of 3D learning methods over 2D.[24]

For physicians, the use of a patient-specific model is preferable to looking at CT and MRI scans as they are viewed on a 2D flat screen.[25] This indicates that 3D-printed models may bring an added aspect to multidisciplinary meeting discussions, perhaps due to the ability to hold and rotate the model from different angles to appreciate dimensions and spatial relationships.

In our study, reconstruction and printing of the 3D heart models required a significant amount of resources and time. The average time for reconstruction and printing was 27±1.7 hours, which may not be practical in a clinical setting. Printers with superior specifications to the Formlabs Form 2 SLA used for this study could be used to decrease printing times, although at a higher initial cost (capital investment for equipment purchase). Alternatively, changing model resolution options may reduce printing times but produce a lower quality model; such

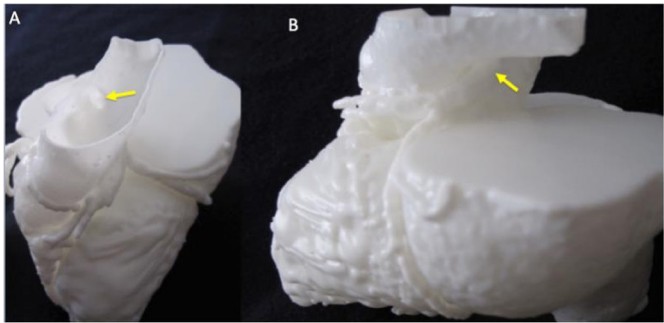

**Figure 4** Detail of ALCAPA model for appreciation of coronary ostium in the pulmonary artery (A) and course of the left coronary artery (B), as indicated by the yellow arrows. ALCAPA, anomalous left coronary artery from pulmonary artery.

changes can be explored further to determine a clinically appropriate balance of quality versus printing speed.

We calculated the average price of production for one model to be £19 GBP (US$25), excluding initial printer purchasing cost. Although we obtained feedback regarding the usefulness of the models, we were unable to establish cost-effectiveness due to the lack of measurable variables such as clinical outcome. Assessment of clinical feasibility relied on feedback from a small group of researchers and clinicians, which may be prone to subjectivity. Further studies to evaluate clinical effectiveness may include measures such as impact of model on clinical outcomes and decision-making. Both clinicians and researchers found the models clear and effective in displaying coronary anatomy and the anomalies, however, clinicians rated the models higher on average. This difference may be explained by the intrinsic differences between the two groups, where clinicians have a stronger background in viewing imaging and anatomical understanding. This is reflected in their self-declared measure of imaging expertise, whereas seven out of nine researchers had only basic imaging expertise. As a result, clinicians likely have a better understanding of coronary artery course and relational structures, and may be able to better appreciate the anatomy displayed by the model. Conversely, the models were still useful for researchers who were unable to fully understand the pathology from CT scans but identified it on the models. These findings suggest that the models are more effectively interpreted by users with a stronger ability to understand relevant anatomy, but still hold purposeful value for individuals with less expertise. Another possible viewing method is the 3D virtual model (eg, 3D pdf files). Although a study by Sun *et al* showed that viewing physical models produced faster and accurate responses than virtual models, they may provide a more cost-friendly alternative than 3D-printed models.[26] This can be explored with comparisons to CT imaging data and physical models in future studies, specifically exploring differences in spatial understanding. Both clinicians and researchers found the models of TGA with abnormal circumflex artery and of ALCAPA the most effective and clearest to understand. Interestingly, comments by three clinicians suggested that the pathology for the ALCAPA case was difficult to see on CT scans, whereas the anatomy was clearly delineated on the 3D-printed models. Consideration of the complexity of the anatomical differences associated with the disease—TGA and ALCAPA being congenital coronary variations with abnormal coronary origin and course—suggest that the two conditions are more difficult to understand and follow than the pathologies of other cases which scored slightly lower. 3D-printed models may potentially confer the most benefit in complex pathologies; this is consistent with the results of a study where anatomical teaching for the three most complicated regions of the body was delivered effectively with 3D modelling techniques.[27]

Segmentation of the models with removal of certain structures was to some extent an operator-dependent procedure. It was, in general, met with good response. Eighty per cent of clinicians and 62% researchers reported that it facilitated viewing the anatomy and the abnormality, while only 5% of clinicians and researchers stated that it made it harder to appreciate the anatomy and abnormality. Two clinicians questioned whether removing parts of the heart was the best approach, as in real life 'we don't have parts removed'. Although no formal review has been done of effectiveness of removing structures during segmentation process, it is a technique that has been used previously.[22 23] One clinician suggested using a more flexible material that enables structures to be bent to view the underlying coronary arteries, and there may be scope in further investigations to trial more flexible materials such as TangoPlus FullCure, which has been used for preoperative surgical training.[27 28] Another option is the use of silicone models. Flexible models can be created with an injection moulding technique using infusion of gel materials such as silicone and urethane into the mould.[29] A more versatile material resulting in bendable structures may allow for preservation of cardiac anatomy and structures, and manual manipulation to view the coronary arteries and anomalies.

One further suggestion by both clinicians and researchers was the use of colour to distinguish coronary arteries from other structures. The colour of certain structures can be 'painted' on the digital file before printing, as demonstrated by McMenamin *et al* who reconstructed prosections of the hand and wrist and highlighted various anatomical features using the software package 3D-Coat (Kompaniya Pilgway Studio, Ukraine). This was shown to create realistic 3D replicas in which 'even small nerves and vessels could be readily distinguished'.[30] The in-house printer used for our study did not support printing with multiple colours, and further studies can be conducted with higher end printers to evaluate the use of colour in looking at coronary arteries anomalies. Awareness of intramurality of anomalous coronary arteries can be immensely useful in planning operations and choosing a suitable surgical strategy. Case 4 (myocardial bridging) was able to demonstrate intramural course which improved stakeholders' understanding from viewing the CT alone, where it may be difficult to delineate. However, although 3D modelling can recreate an intramural CAA course, it does not add to data available from CT imaging. It is hence more suitable to be used for improving understanding or planning surgical approaches, rather than confirming intramurality.

The cases for this study were selected from a hospital database, thus, the overall representation of CAAs in this study may be skewed towards more symptomatically severe cases requiring medical attention. Our results may therefore not be generalisable to CAAs of all severities. However, uncomplicated cases of CAA not necessitating treatment would likely not benefit significantly from a 3D-printed model. Our study was limited by the relatively small number of cases. Patients were selected by cardiac radiology consultants to provide a range of different CAAs

at different stages, for example, clarifying anomaly, incidental finding, follow-up. Although a spectrum of CAAs was successfully produced, other conditions such as atresia and duplication were not assessed due to limited general population incidence. An appreciation of the significant haemodynamic and structural differences of the different anomalies suggests that further studies looking at other anomalies may need to be done to extrapolate our results to varying classifications of the anomalies.[16] Furthermore, it would be interesting to explore differences between subgroups, particularly between clinicians with and without imaging expertise, whereas in this study this was limited by the small sample size of participants and the focus was comparing participants with and without medical training. Wider future use of 3D printing in CAAs may require a more standardised protocol in production and quality assessment. Independent review of the models and accuracy to scan was done in this study, showing accurate reconstruction of the models and coronary arteries. A formal method of assessment such as Bland-Altman analysis—quantifying agreement between the dimensions of CT scans and 3D models—has been used to confirm accuracy of the 3D-printed models.[31 32] These can be used in the future to verify consistency.

## CONCLUSION

Where small discrepancies in coronary anatomy can be significant in determining prognosis and presentation, it is important that the exact anatomy should be fully understood before management decisions can be appropriately made. 3D-printed heart models can be feasibly used to recreate coronary artery anatomy and enhance understanding of the abnormality. The models could complement CT scans to display anatomy and abnormalities, and in more anatomically complex cases display structural relationships not entirely clear on the scans for users with moderate or no expertise in viewing imaging data. Future studies towards the use of 3D printing in CAAs should investigate the best purpose of use of 3D-printed models within the spectrum of CAAs, in terms of severity and classification. There may be scope for use in preoperative planning and decision-making, as well as teaching and clinical consultations. Cost–benefit should be determined before its role can be established alongside conventional imaging alternatives for investigating CAAs such as CT angiography.

**Contributors** ML created the models and carried out the survey, with RB, as well as analysing survey results. SM-E, MH, NM, EGM, CB-D and MC aided with selection of cases, clinical insight and model checking. GB conceived the study and oversaw the creation of models and the analysis. ML drafted the manuscript with GB and all authors critically contributed to revising and finalising the manuscript.

**Funding** The work is supported by a David Telling Charitable Trust equipment grant and by the British Heart Foundation. The authors also acknowledge support from the Bristol National Institute for Health Research (NIHR) Biomedical Research Centre.

**Disclaimer** The views expressed in this publication are those of the authors and not necessarily those of the NHS, the National Institute for Health Research or the Department of Health.

**Competing interests** None declared.

**Patient consent for publication** Not required.

**Provenance and peer review** Not commissioned; externally peer reviewed.

**Data sharing statement** All data pertinent to the study are included in the manuscript, no additional unpublished data.

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
