## [Reviewer comments · BMJ Open]

ARTICLE DETAILS

TITLE (PROVISIONAL)	Evaluating 3D printed models of coronary anomalies: a survey amongst clinicians and researchers at a University Hospital in the United Kingdom
AUTHORS	Lee, Matthew; Moharem-Elgamal, Sarah; Beckingham, Rylan; Hamilton, Mark; Manghat, Nathan; Milano, Elena; Bucciarelli Ducci, Chiara; Caputo, Massimo; Biglino, Giovanni

VERSION 1 – REVIEW

REVIEWER	Joseph Vettukattil Helen DeVos Childrens Hospital of Spectrum Health, USA
REVIEW RETURNED	28-Jul-2018

GENERAL COMMENTS	The study explores the feasibility of 3D printed models of coronary anomalies detected on CT angiograms and scored the potential for its clinical application through questionnaires to researchers and clinicians. The study methodology is well constructed and executed. As an clinician imaging person, after reading the article I am not convinced about the clinical utility of 3D printing compared to 2D reconstruction of the coronaries from CT. I have a few questions to the authors: 1. The feasibility of 3D printing from CT has already been established hence the first objective of the study is already been met.2. The task of assessing clinical utility is difficult without evidence of clinical outcomes or interventions. Without comparing the clinical management plan or outcome the impact of 3D printing on clinical use is very subjective. Here the attempt to answer the clinical use is made with a group of researchers and with clinicians of expertise.3. Though the case selection parameters are given, out of how many cases the 7 were selected and why some significant anomalies like anomalous course of the coronary arteries not included is unclear. Hence one of the objectives is not fully met by this study.4. Was a color 3D reconstruction of the CT used for the comparison or was it with 2D MPR?
--

REVIEWER	Andreas Giannopoulos University Hospital Zurich, Zurich, Switzerland
REVIEW RETURNED	30-Sep-2018

GENERAL COMMENTS	I had the pleasure to Review the manuscript by Lee et al on the value of 3D Printing in complex coronary anomalies. Although the number of the subjects was limited, the authors were able to
---

	demonstrate the added multidisciplinary value in a subjective and objective manner. I feel that the manuscript could be enhanced with more variants of coronary anomaly, given that the authors could have access to such cases. Other than that a graphical illustration of the study design could be of aid easing the reader. In this reviewer's opinion the cost-efficiency of printing of such models is important to elucidate.
--	---

REVIEWER	Aurelio Secinaro Bambino Gesù Children's Hospital and Research Institute, Rome, Italy
REVIEW RETURNED	25-Oct-2018

GENERAL COMMENTS	Faster 3D prototyping time can be achieved with other 3D printing technology than Form2. Please, can you add comments about other possible 3D printing strategies that can fit this specific clinical application. Did you provide Clinicians and Researchers also 3D virtual model (3D PDF file or Volume Rendering Reconstructions) for comparison? It is debated that they be equally useful for a better understanding of the 3D anatomy, I suggest to comment about this in the discussion. One of the challenging question in assessing coronary artery abnormalities is the ability to recognise coronary intramural course. The ALCAPA case for example looks suspicious. This might be anecdotal but in case you have a surgical confirmation of the detailed anatomy it would be interesting to have comment about the additional value of 3D printed model in raising suspicion of intramural.
--

VERSION 1 – AUTHOR RESPONSE

Reviewer(s)' Comments to Author:

Reviewer: 1

Reviewer Name: Joseph Vettukattil

Institution and Country: Helen DeVos Childrens Hospital of Spectrum Health, USA

Please state any competing interests or state 'None declared': NIL

Please leave your comments for the authors below

The study explores the feasibility of 3D printed models of coronary anomalies detected on CT angiograms and scored the potential for its clinical application through questionnaires to researchers and clinicians. The study methodology is well constructed and executed. As a clinician imaging person, after reading the article I am not convinced about the clinical utility of 3D printing compared to 2D reconstruction of the coronaries from CT.

I have a few questions to the authors:

1. The feasibility of 3D printing from CT has already been established hence the first objective of the study is already been met.

We agree that the use of CT imaging in 3D printing has been established. The manuscript has been edited to mention this.

Our aim was to assess suitability of CT imaging specifically for printing coronary artery anomalies, including possible issues related to segmentation. As far as we are aware, no studies have shown printing feasibility for coronary artery anomalies specifically. Furthermore, coronary arteries may be difficult to follow due to adjacent structures; we present a method of removing certain parts of cardiovascular anatomy to reveal the coronary course more clearly, as mentioned in our 'Materials and Methods' section. This removal takes place during the CT segmentation phase.

Manuscript (Introduction): The use of CT imaging within 3D printing in medicine has been widely established.

2. The task of assessing clinical utility is difficult without evidence of clinical outcomes or interventions. Without comparing the clinical management plan or outcome the impact of 3D printing on clinical use is very subjective. Here the attempt to answer the clinical use is made with a group of researchers and with clinicians of expertise.

Many thanks for your valuable comments. We have added a section mentioning possible subjectivity in measuring clinical feasibility. Due to the retrospective nature of our study, we were unable to assess the impact of the model on clinical outcome. However, although clinical outcomes and impact on intervention and management plans are undoubtedly important measures of effectiveness, we believe other factors such as communication and understanding to be equally important, particularly in congenital cardiac disease where outcome is multifactorial.

Manuscript (Discussion): Assessment of clinical feasibility relied on feedback from a small group of researchers and clinicians, which may be prone to subjectivity. Further studies to evaluate clinical effectiveness may include measures such as impact of model on clinical outcomes and decision-making.

3. Though the case selection parameters are given, out of how many cases the 7 were selected and why some significant anomalies like anomalous course of the coronary arteries not included is unclear. Hence one of the objectives is not fully met by this study.

We have edited the manuscript to further clarify the details of the selection process, thanks for highlighting this.

Manuscript (Materials and Methods: Case Selection): Two cardiac radiology consultants with >15 years' experience in cardiac CT (NM, MH) selected n=7 cases from our Centre's database of roughly 2,000 patients having coronary CT over the last 10 years. [...] Cases were selected to best represent a spectrum of CAAs at different incidences and timelines.

Manuscript (Discussion): Patients were selected by cardiac radiology consultants to provide a range of different CAAs at different stages e.g. clarifying anomaly, incidental finding, follow-up. Although a spectrum of coronary artery anomalies was successfully produced, other conditions such as atresia and duplication were not assessed due to limited general population incidence.

4. Was a color 3D reconstruction of the CT used for the comparison or was it with 2D MPR ?

2D MPR was used when presenting in comparison to the 3D printed model. Colour delineation on the CT segmentation of the reconstruction was removed to create a more realistic experience of viewing the scans. A 3D reconstruction of the CT ('virtual model') was not provided.

Reviewer: 2

Reviewer Name: Andreas Giannopoulos

Institution and Country: University Hospital Zurich, Zurich, Switzerland

Please state any competing interests or state 'None declared': None Declared

Please leave your comments for the authors below

I had the pleasure to Review the manuscript by Lee et al on the value of 3D Printing in complex coronary anomalies. Although the number of the subjects was limited, the authors were able to demonstrate the added multidisciplinary value in a subjective and objective manner. I feel that the manuscript could be enhanced with more variants of coronary anomaly, given that the authors could have access to such cases.

Thank you for your comments. We agree that more variants of coronary anomaly would add value to overall evaluation. The cases were chosen to best represent a spectrum of CAAs at different incidences and timelines, as the first study to evaluate 3D printing for coronary arteries. Several anomalies were unable to be assessed due to limited incidence within our database. We have revised our manuscript to include the mentioned details. Further studies in the future could assess models presenting more variants of coronary anomalies.

Manuscript (Materials and Methods: Case Selection): Cases were selected to best represent a spectrum of CAAs at different incidences and timelines.

Manuscript (Discussion): Patients were selected by cardiac radiology consultants to provide a range of different CAAs at different stages e.g. clarifying anomaly, incidental finding, follow up. Although a spectrum of coronary artery anomalies was successfully produced, other conditions such as atresia and duplication were not assessed due to limited general population incidence.

Other than that a graphical illustration of the study design could be of aid easing the reader.

We have added an illustration of the study design to make it easier for readers.

In this reviewer's opinion the **cost-efficiency of printing** of such models is important to elucidate.

A description has been added discussing cost-efficiency of 3D printing:

Manuscript (Discussion): We calculated the average price of production for one model to be £19 GBP (\$25 USD), excluding initial printer purchasing cost. Although we obtained feedback regarding the usefulness of the models, we were unable to establish cost-effectiveness due to the lack of measurable variables such as clinical outcome. Assessment of clinical feasibility relied on feedback from a small group of researchers and clinicians, which may be prone to subjectivity. Further studies to evaluate clinical effectiveness may include measures such as impact of model on clinical outcomes and decision-making.

Reviewer: 3

Reviewer Name: Aurelio Secinaro

Institution and Country: Bambino Gesù Children's Hospital and Research Institute, Rome, Italy

Please state any competing interests or state 'None declared': None declared

Please leave your comments for the authors below

Faster 3D prototyping time can be achieved with other 3D printing technology than Form2. Please, can you add comments about other possible 3D printing strategies that can fit this specific clinical application.

Thank you for your valuable comment. We have edited the manuscript to include discussion regarding the possibility of alternative printing methods.

Manuscript (Discussion): Printers with superior specifications to the Formlabs Form 2 SLA used for this study could be utilised to decrease printing times, albeit at a higher initial cost (capital investment for equipment purchase). Alternatively, changing model resolution options may reduce printing times but produce a lower quality model; such changes can be explored further to determine a clinically appropriate balance of quality versus printing speed.

Did you provide Clinicians and Researchers also 3D virtual model (3D PDF file or Volume Rendering Reconstructions) for comparison? It is debated that they be equally useful for a better understanding of the 3D anatomy, I suggest to comment about this in the discussion.

Clinicians and researchers were only provided with 2D MPR as well as the model to manipulate for viewing in their hands. A section discussing the use of 3D virtual models has been added:

Manuscript (Discussion): Another possible viewing method is the 3D virtual model (e.g. 3D pdf files). Although a study by Sun et al. showed that viewing physical models produced faster and accurate responses than virtual models, they may provide a more cost-friendly alternative than 3D printed models.^[26] This can be explored with comparisons to CT imaging data and physical models in future studies, specifically exploring differences in spatial understanding.

One of the challenging question in assessing coronary artery abnormalities is the ability to recognise coronary intramural course. The ALCAPA case for example looks suspicious. This might be anecdotal but in case you have a surgical confirmation of the detailed anatomy it would be interesting to have comment about the additional value of 3D printed model in raising suspicion of intramurality.

Thank you for the insightful comment. We have added a section in the manuscript discussing the use of the models for intramurality - Case 4 (myocardial bridging) showed an intramural course of coronary artery which was well received by stakeholders. However, as 3D models are derived from CT imaging and unable to augment/intensify such abnormalities, they may not be suitable for confirming intramurality.

Manuscript (Discussion): Awareness of intramurality of anomalous coronary arteries can be immensely useful in planning operations and choosing a suitable surgical strategy. Case 4 (myocardial bridging) was able to demonstrate intramural course which improved stakeholders' understanding from viewing the CT alone, where it may be difficult to delineate. However, although 3D modelling can recreate an intramural CAA course, it does not add to data available from CT imaging. It is hence more suitable to be used for improving understanding or planning surgical approaches, rather than confirming intramurality.

VERSION 2 – REVIEW

REVIEWER	Andreas Giannopoulos University Hospital Zurich
REVIEW RETURNED	08-Dec-2018
GENERAL COMMENTS	The authors responded to all of my comments appropriately. I have no further remarks.
REVIEWER	Aurelio Secinaro Bambino Gesù Children's Hospital
REVIEW RETURNED	01-Jan-2019

GENERAL COMMENTS	Dear authors, Thanks for changes applied to the original manuscript. I have no other improvements to suggest.
---

VERSION 2 – AUTHOR RESPONSE

We are pleased that the Reviewers found the article improved and suitable for publication.

We have addressed the comment of the Editor by revising the title of the manuscript including the research setting (University Hospital in the United Kingdom). The Abstract has also been updated accordingly.

Addition to manuscript (Updated title): Evaluating 3D printed models of coronary anomalies: a survey amongst clinicians and researchers at a University Hospital in the United Kingdom

Addition to manuscript (Abstract): Setting: University Hospital (division of cardiology) in the United Kingdom.